# ADVEDM: Fine-grained Adversarial Attack against VLM-based Embodied Agents

**Yichen Wang**[1,2,4,5,6]    **Hangtao Zhang**[6]    **Hewen Pan**[1,2,4,5,6]    **Ziqi Zhou**[1,2,3,7]

**Xianlong Wang**[8]    **Peijin Guo**[1,2,4,5,6]    **Lulu Xue**[1,2,4,5,6]    **Shengshan Hu**[1,2,4,5,6]

**Minghui Li**[9]    **Leo Yu Zhang**[10]

[1]National Engineering Research Center for Big Data Technology and System
[2]Services Computing Technology and System Lab    [3]Cluster and Grid Computing Lab
[4]Hubei Engineering Research Center on Big Data Security
[5]Hubei Key Laboratory of Distributed System Security
[6]School of Cyber Science and Engineering, Huazhong University of Science and Technology
[7]School of Computer Science and Technology, Huazhong University of Science and Technology
[8]Department of Computer Science, City University of Hong Kong
[9]School of Software Engineering, Huazhong University of Science and Technology
[10] School of Information and Communication Technology, Griffith University
{wangyichen, hangt_zhang, hewenpan, zhouziqi, gpj, lluxue, hushengshan, minghuili}@hust.edu.cn
xianlong.wang@my.cityu.edu.hk, leo.zhang@griffith.edu.au

## Abstract

Vision-Language Models (VLMs), with their strong reasoning and planning capabilities, are widely used in embodied decision-making (EDM) tasks in embodied agents, such as autonomous driving and robotic manipulation. Recent research has increasingly explored adversarial attacks on VLMs to reveal their vulnerabilities. However, these attacks either rely on overly strong assumptions, requiring full knowledge of the victim VLM, which is impractical for attacking VLM-based agents, or exhibit limited effectiveness. The latter stems from disrupting most semantic information in the image, which leads to a misalignment between the perception and the task context defined by system prompts. This inconsistency interrupts the VLM's reasoning process, resulting in invalid outputs that fail to affect interactions in the physical world. To this end, we propose a fine-grained adversarial attack framework, ADVEDM, which modifies the VLM's perception of only a few key objects while preserving the semantics of the remaining regions. This attack effectively reduces conflicts with the task context, making VLMs output valid but incorrect decisions and affecting the actions of agents, thus posing a more substantial safety threat in the physical world. We design two variants of based on this framework, ADVEDM-R and ADVEDM-A, which respectively remove the semantics of a specific object from the image and add the semantics of a new object into the image. The experimental results in both general scenarios and EDM tasks demonstrate fine-grained control and excellent attack performance.

## 1 Introduction

Visual-language models (VLMs) such as GPT-4 [1] and Gemini-2.0 [2] have been widely adopted for embodied decision-making (EDM) tasks in embodied agents, including autonomous driving [3, 4, 5] and robotic manipulation [6, 7, 8], due to their powerful reasoning and planning capabilities. In these tasks, VLMs generate decisions and plannings based on current inputs and system states, and

39th Conference on Neural Information Processing Systems (NeurIPS 2025).

then convert them to control code to guide physical entities (*e.g.*, vehicles, robotic arms) in their interactions with the real world.

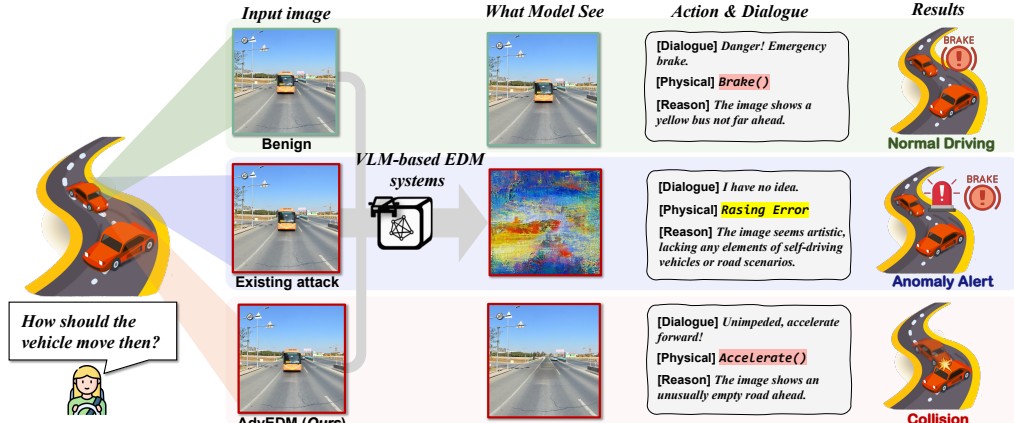

Figure 1: Comparison of our attack framework with existing works in attacking VLM-based agents. Existing attacks disrupt most of the semantics in the original image, causing the VLM to generate invalid responses. In contrast, our attack selectively alters the VLM's perception of a specific object while preserving the semantic integrity of other regions. As a result, the VLM produces valid yet incorrect decisions, effectively influencing the system's interaction with the physical world.

However, VLMs have shown vulnerability to adversarial attacks [9, 10, 11, 12, 13, 14], where the adversary manipulates the model's output by introducing imperceptible perturbations into input images. Existing attacks on VLMs can be classified into two categories: white-box attacks and black-box attacks. White-box attacks, such as AttackBard [15] and CroPA [16], generate adversarial examples by optimizing the end-to-end process, directly manipulating the textual outputs. But such attacks are impractical for VLM-based embodied agents, as it is difficult for the adversary to access the LLM modules within the VLM, which are fine-tuned on proprietary datasets for specific tasks [17].

In contrast, black-box attacks where the adversary's knowledge of the victim model is limited are more practical. However, a fully black-box setting poses significant challenges for attackers, incurring significant computational costs and limited attack effectiveness [18]. Therefore, many existing works (like AttackVLM [10], CLIP-based Attack [19] and VT-Attack [20]) propose a compromise setting, where the attacker has access to only the vision-text encoder of VLMs, referred to as the gray-box setting. Since the vision-text encoder in VLM-based EDM systems is directly used in its pre-trained form [21] and easy for the adversary to obtain, we focus on the gray-box attacks in this paper. These attacks introduce adversarial perturbations to move the image's embedding away from the clean image's embedding, thereby disrupting the VLM's perception of the image. However, such attacks have limited effectiveness against the VLM-based EDM system in embodied agents, which only make the system output **invalid results without guiding entity's action in the physical world** (like error reports). This is because these systems employ chain-of-thought (CoT) techniques [22] to perform reasoning and task planning. The CoT first analyzes the system's current state and perceived inputs, then proceeds to further reasoning based on the task description provided by system prompts [23, 24]. Existing attacks of this type alter the system's overall perception of the image, causing a conflict with the system prompts' description, thereby interrupting the reasoning process and resulting in invalid outputs rather than decisions and plannings. This process is illustrated in Fig. 1.

In this paper, we propose a novel fine-grained adversarial attack framework, ADVEDM, which makes the VLM-based EDM system output valid but incorrect decisions. As illustrated in Fig. 1, our attack disrupts the CoT in the VLM-based EDM system by modifying the VLM's perception of the existence of several key objects while retaining the original semantics of other parts. This significantly reduces conflicts with the task context, ensuring the integrity and logical consistency during the reasoning process. Consequently, the system outputs valid but incorrect decisions that can alter the entity's action, leading to a more substantial safety threat in the physical world.

Specifically, we design two attack methods based on this framework, ADVEDM-R and ADVEDM-A, which respectively remove the semantics of an object from the image and add the semantics of a

new object into the image, while preserving the semantics of others. The implementation of these attacks faces two technical challenges: first, how to select appropriate regions in the image for the removal and addition of the target object's semantics; and second, how to preserve the semantics of other regions after modifying the target semantics. To address the first challenge, we propose a selection strategy based on the similarities between cross-modality embeddings, which leverages the correlation between the VLM's vision and text encoders [25, 26] by calculating the similarity between image patch token embeddings and object text embeddings. For the second challenge, considering that the image embeddings are typically generated by the attention mechanism of ViT [27], we propose attention-$[patch]$ fixation, which preserves the semantics of the remaining parts by maintaining the product of the attention weights and patch token embeddings.

We evaluate our methods in both general image description scenarios and two representative embodied decision-making tasks: autonomous driving and robotic manipulation. In general scenario, the average attack success rates (ASR) for the two variants are 76.8% and 70.2%, with semantic preservation rates of 66.7% and 71.6%. In EDM tasks, our method achieved an attack success rate of over 70% and 64% in autonomous driving and robotic manipulation respectively, significantly outperforming existing attacks. These results highlight the excellent fine-grained control of our attacks and their effectiveness in posing a real safety threat to VLM-based EDM systems. More demos of our attacks in real-world scenarios can be found on our website https://advedm.github.io/.

In conclusion, the contribution of this paper can be summarized as follows: (1) We propose a novel fine-grained adversarial attack framework ADVEDM in the gray-box setting that selectively modifies the semantics of key objects perceived by VLM-based EDM systems, disrupting their reasoning process and leading to valid but incorrect decisions, thus increasing real-world safety risks. This aims to reveal the vulnerabilities of current VLM-based EDM systems and foster future efforts to enhance their robustness. (2) Based on the framework, we design two attacks ADVEDM-R and ADVEDM-A, which respectively remove the semantics of a specific object or add the semantics of a new object to the image. (3) The experimental results in both general scenarios and embodied decision-making tasks indicate the excellent fine-grained control and effectiveness of our attacks.

## 2 Related Work

### 2.1 VLM-Based Embodied Decision-Making System

Due to their exceptional logical reasoning capabilities, VLMs have been widely applied to embodied decision-making tasks [28, 29, 30]. The Chain-of-Thought (CoT) is widely used in VLM-based EDM systems [22, 23], which breaks down the task into logical steps, refining the model's decision-making based on the current context and task requirements. Two prominent applications are autonomous driving and robotic manipulation. In autonomous driving, VLMs fine-tuned on specialized datasets, such as DriveLM [4], Dolphins [21], and DriveGPT [31], process road information captured by sensors like cameras. VLMs in this task, combined with predefined system prompts, enable real-time planning and adjustments to the vehicle's driving state through CoT reasoning process. In the robotic manipulation task, VLMs first perceive the input visual images, then combine them with received instructions and system prompts to perform reasoning through CoT and generate decisions and plannings regarding the robot's actions like rotation, movement, and grasping. Finally, the post-processing module translates these decisions into control code to manipulate its interactions with the physical world [32]. In conclusion, VLM-based EDM systems play a crucial role in embodied AI tasks. While extensive research has been conducted on the robustness of VLMs themselves [10], the robustness and security of VLM-based EDM systems remain unexplored.

### 2.2 Adversarial Attack against VLMs

Adversarial attack involves manipulating the outputs of models by introducing imperceptible perturbations to the inputs [33, 34, 35, 36, 37, 13, 38, 39, 40]. With the widespread application of VLMs, there have been increasing researches focused on attacks against VLMs in recent years. Most of existing works [19, 41, 42, 43, 44] focus on designing attack methods in a white-box setting, where it is assumed that the adversary has full access to the victim model and other relevant information to launch attack. These methods typically optimize the adversarial noise by minimizing the difference between the logits of probability of the outputs and the pre-defined target text, thereby enabling end-to-end attacks. Despite their high attack success rates, these attacks are impractical for the

VLM-based EDM system, as it is difficult for the adversary to access VLMs fine-tuned on various datasets for specific tasks [45].

To this end, some works proposed attacks in more general scenarios [9, 15, 46, 10], where the adversary has limited knowledge about the victim VLMs. They usually employ pre-trained vision-text encoders of VLMs as surrogate models, such as CLIP [47, 48]. These attacks involve making the embeddings of adversarial examples diverge from those of original images or closer to those of target images to disrupt the perception of VLMs. Although they are more practical for attacking VLM-based EDM systems whose vision-text encoders are usually pre-trained and easy to obtain, their effectiveness is limited due to the lack of fine-grained control. Specifically, they disrupt most of the semantics perceived by the VLM from original images, thus interrupting the reasoning process of VLMs and leading to invalid results that fail to influence interactions with the physical world. In this work, we design two fine-grained adversarial attack methods ADVEDM-R and ADVEDM-A that disrupt the perception of VLMs by precisely removing or adding the semantics of target objects, while preserving other semantics in the original image.

# 3 Preliminary

## 3.1 Background

Following existing works [49, 28], the VLM-based EDM system consists of two components, as shown in Fig. 2. The first component is the decision-making module implemented by a VLM, which includes a pre-trained vision-text encoder (*e.g.*, CLIP [25]) and a fine-tuned LLM. The vision-text encoder encodes the environmental image along with the received textual instructions and system prompts, as described in Eq. 1.

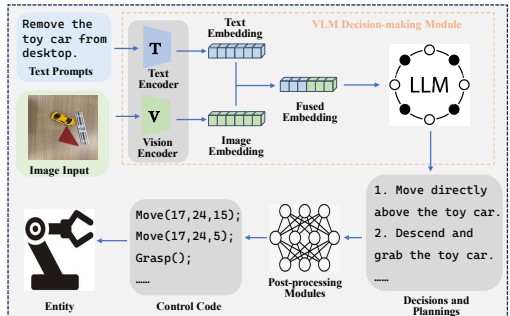

$$\phi_i = E_v(I), \quad \phi_t = E_t(T) \qquad (1)$$

where $E_v(\cdot)$ and $E_t(\cdot)$ are the vision encoder and text encoder, while $\phi_i$ and $\phi_t$ are the em-

Figure 2: The framework of VLM-based embodied decision-making system.

beddings of the input image $I$ and text instruction $T$. Note that the image encoders in VLMs are typically based on the Transformer architecture, where the image embeddings $\phi_i$ include a [CLS] token embedding (representing overall semantics) and patch token embeddings (representing local semantics) [50]. We denote them as $[cls]$ and $[patch]$.

The LLM in the decision-making module is fine-tuned on a proprietary task-specific dataset. It takes as input the fused and concatenated embeddings, performs reasoning through CoT, and generates decisions and plannings. This process is formulated as $T_D = \mathcal{M}(\phi_i, \phi_t; \theta)$, where $\mathcal{M}$ is the LLM with parameters $\theta$, and $T_D$ is the textual outputs of decisions and plannings.

Upon generating decisions and plannings, the post-processing module translates and converts them into executable control code for operating physical hardware of the entity, which can be expressed as $C = f_p(T_D)$. $C$ is the control code and $f_p$ represents the post-processing module.

## 3.2 Threat Model

**Adversary's goal.** To achieve fine-grained attack effectiveness, the adversary's goal as inducing the textual decisions where only the content regarding the target object is altered, while the rest remain unchanged. Here, we formally define our fine-grained adversarial attack. Specifically, we decompose the decision $T_D$ into descriptions for $n$ individual objects in the image, which can be expressed as $T_D = \{D_{obj_1}, D_{obj_1}, ..., D_{obj_n}\}$. Then the attack is formulated as Eq. (2).

$$
\begin{aligned}
\min \quad & \mathrm{Sim}(D'_{obj_t}, D_{obj_t}) \\
\text{s.t.} \quad & \mathrm{Sim}(D'_{obj_i}, D_{obj_i}) > \delta, \quad i \neq t \\
& \|I' - I\|_2 < \epsilon
\end{aligned}
\qquad (2)
$$

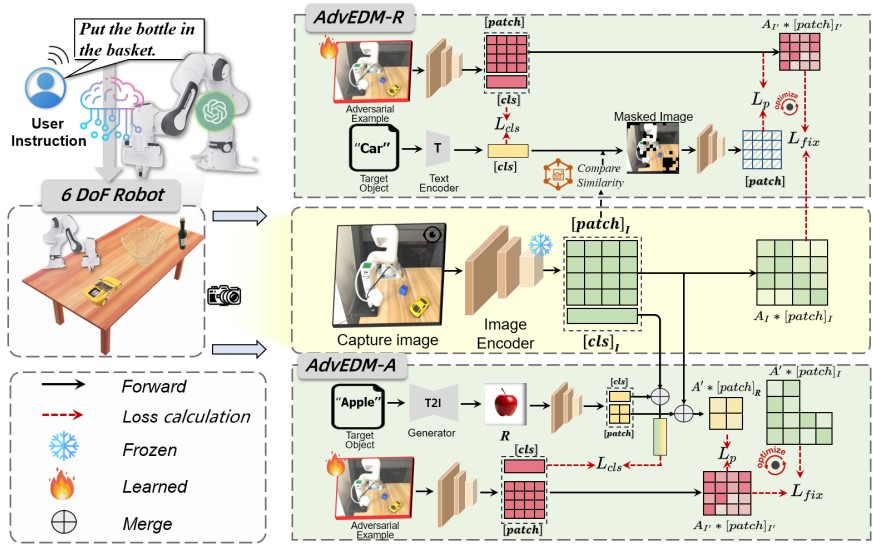

Figure 3: The pipeline of our methods ADVEDM-R and ADVEDM-A.

where $obj_t$ is the target object, and $D'_{obj_t}$ and $D'_{obj_i}$ are decisions generated under adversarial examples $I'$, while $D_{obj_t}$ and $D_{obj_i}$ are under clean inputs $I$. $\text{Sim}(\cdot)$ measures semantic similarity between texts, and $\delta$ and $\epsilon$ represent the constraints for the semantic preservation and visual stealthiness.

**Adversary's capacity.** Since the textual input $T$ is usually pre-defined like system prompts [6, 24] and difficult to manipulate from from external sources, we assume the adversary can only perturb the image inputs to launch an attack. Following many previous works like [10, 20, 19], we design our attacks in the gray-box setting, where the adversary can only utilize the vision-text encoder of VLMs as surrogate models to generate adversarial examples. Moreover, we also extend our attacks to the black-box setting and provide a detailed discussion in Appendix D.

## 4 Methodology

### 4.1 Intuition

Unlike previous adversarial attacks on VLMs in general tasks like VQA, effectively attacking the VLM-based EDM system requires inducing valid yet incorrect decisions that impact real-world interactions. This necessitates a fine-grained adversarial attack that selectively modifies the target object's semantics while preserving the overall reasoning integrity.

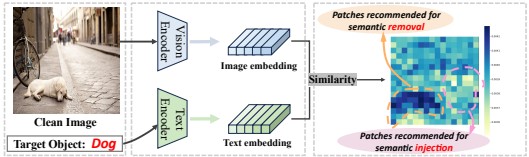

Figure 4: Our strategy to select the target regions for semantic removal and semantic addition.

Specifically, we propose two feasible schemes: (1) **Semantic Removal (SR)**, namely removing the semantics of a specific object while preserving those of others, and (2) **Semantic Addition (SA)**, which involves adding the semantics of a specified object without altering the semantics of others. For SR, the target output $T'_D$ should exclude the semantics of the target object $obj_t$ while preserving those of the remaining $n-1$ objects. For SA, it should incorporate both the original $n$ objects and $obj_t$. Both schemes face two challenges: first, how to identify the key regions in the image for SR or SA to ensure the effectiveness of attacks, and second, how to maintain the semantics of the remaining parts of the image.

Given that the vision-text encoder in most VLMs is built on the Transformer architecture [45, 48], we tackle the first challenge by identifying key regions based on the similarity between image patch token embeddings and the text embedding of $obj_t$. This similarity effectively quantifies the extent to which each patch contains the target object's semantics, due to the ability of the vision-text encoder to align visual and textual semantics. As shown in Fig. 4, patches with higher similarity scores contain

more relevant semantic information. For SR, we choose the patches with higher similarity to the text embeddings of target objects and erase their semantics. For SA, we select contiguous background regions with relatively lower similarity to foreground objects' embeddings, ensuring the injected semantics minimally influence the semantics of other objects.

As for the second challenge, we adopt an attention-$[patch]$ fixation approach. Specifically, during the optimization of adversarial perturbation, we ensure that the product of attention weights and patch token embeddings for the remaining regions closely matches that of the original image. This approach effectively preserves both the overall semantics and local detailed features of them [51].

## 4.2 Our Methods

According to our intuition, we propose two attack methods ADVEDM-R and ADVEDM-A, which remove the semantics of a specific object from the image and inject the semantics of a new object.

### 4.2.1 ADVEDM-R

We first demonstrate how to identify regions with target object's semantics in the image. We calculate the cosine similarity between the patch token embeddings and the text embedding of target object, and then mark the patches with higher similarity and form a mask. This process is formally described in Eq. 3 and 4.

$$S = CS([patch]_I, E_t(T_{tar}))  \tag{3}$$

$$mask_i = \left\{ \begin{array}{ll} 0, & \text{if } s_i > \xi \\ 1, & \text{if } s_i \leq \xi \end{array} \right\}  \tag{4}$$

where $[patch]_I$ is the patch token embeddings of clean image $I$, $E_t(T_{tar})$ is the text embedding of the target object encoded by $E_t$ and $CS(\cdot)$ is the cosine similarity function. The similarity vector $S \in \mathbb{R}^{n \times 1}$, where $n$ is the number of image patches. $mask$ is also an $n$-dimensional vector, whose $i$-th element $mask_i$ is determined by comparing the corresponding element $s_i$ in $S$ with a predefined threshold $\xi$. Elements in $mask$ with a value of 0 indicate that the corresponding image patches contain richer semantics of the target object.

After obtain the $mask$, we remove the semantic of target object from both global and local perspectives. For global semantics, we push the [CLS] token embedding that represents the overall semantics of the image, away from $E_t(T_{tar})$ to remove the target object's global semantics, as shown in Eq. 5.

$$\mathcal{L}_{cls} = CS([cls]_{I'}, E_t(T_{tar}))  \tag{5}$$

For local semantics, we utilize the obtained $mask$ to erase the patches containing target semantics from the image, and denote the masked image as $M$. In $M$, the patches originally rich in semantics of the target object are replaced by meaningless 0-pixel values. Then, we align the embeddings of corresponding patches in the adversarial example with those in $M$. This process is shown in Eq. 6.

$$\mathcal{L}_p = -(1 - mask) * CS([patch]_{I'}, [patch]_M)  \tag{6}$$

Additionally, we propose the attention-$[patch]$ fixation, ensuring that the key features of the rest parts remain consistent with those of the original image, as shown in Eq. 7.

$$\mathcal{L}_{fix} = -mask * CS(A_{I'} * [patch]_{I'}, A_I * [patch]_I)  \tag{7}$$

where $A_I$ and $A_{I'}$ are the means of attention weights across all attention layers in $E_v$ of the original image and adversarial example, as they reflect the semantic significance of each patch [51].

In conclusion, the entire process of ADVEDM-R can be expressed as Eq. 8, and $w_1$ to $w_3$ are the weights of each loss item. The pipeline of ADVEDM-R is shown in Fig. 3.

$$\min_{I'} w_1 * \mathcal{L}_{cls} + w_2 * \mathcal{L}_p + w_3 * \mathcal{L}_{fix}$$
$$s.t. \quad \|I' - I\|_2 < \epsilon  \tag{8}$$

Table 1: Quantitative results of attacks on MS-COCO dataset in the image description task.

| Models / Attacks | LLAVA-v2 | | | MiniGPT4 | | | Otter-Image | | | BLIP-2 | | | OFLMG-v2 | | | Average | | |
|---|---|---|---|---|---|---|---|---|---|---|---|---|---|---|---|---|---|---|
| | ASR(%) | SPR(%) | SS | ASR(%) | SPR(%) | SS | ASR(%) | SPR(%) | SS | ASR(%) | SPR(%) | SS | ASR(%) | SPR(%) | SS | ASR(%) | SPR(%) | SS |
| PGD | 71.7 | 22.5 | 0.390 | 70.8 | 17.9 | 0.313 | 73.4 | 18.1 | 0.363 | 74.5 | 17.0 | 0.426 | 79.2 | 20.4 | 0.350 | 73.9 | 19.2 | 0.368 |
| MF-it | 81.8 | 19.5 | 0.343 | 77.3 | 13.7 | 0.280 | **87.5** | 13.8 | 0.290 | 85.0 | 11.1 | 0.340 | 84.1 | 15.9 | 0.290 | 83.1 | 14.8 | 0.309 |
| MF-ii | **83.6** | 15.0 | 0.363 | **82.4** | 9.00 | 0.252 | 87.3 | 10.3 | 0.224 | **85.6** | 10.9 | 0.355 | **86.9** | 11.1 | 0.242 | **85.2** | 11.3 | 0.287 |
| ADVEDM-R | 75.2 | **71.3** | **0.705** | 73.9 | **66.4** | **0.626** | 80.9 | **64.4** | **0.652** | 74.3 | **65.3** | **0.737** | 79.6 | **66.2** | **0.695** | 76.8 | **66.7** | **0.683** |
| ADVEDM-A | 68.4 | **75.1** | **0.758** | 67.1 | **72.6** | **0.657** | 72.8 | **69.4** | **0.728** | 69.5 | **68.2** | **0.756** | 73.3 | **72.9** | **0.732** | 70.2 | **71.6** | **0.726** |

### 4.2.2 ADVEDM-A

In the implementation of ADVEDM-A, we first manually select contiguous patches of size $m \times m$, typically from the background or areas containing minimal specific objects. The selection can refer patches with lower embedding similarity to the foreground objects' text embeddings. After the selection, we apply the same procedure as in Eq. 4 to mark the selected patches as 0 and the remaining as 1, thereby generating the corresponding $mask$.

Due to the modality gap between image and text inputs in image-text encoders [52], textual descriptions of the target object cannot effectively inject semantics into the image embeddings, especially the patch token embeddings. To address this, we utilize a reference image $R$ with $m \times m$ patches that solely contains the target object (generated by a text2image model [53]). Here, we also incorporate the target semantics into the original image from both global and local perspectives. For global semantics, we align the $[cls]$ of the adversarial example with the weighted fusion of $[cls]$ from the clean and target images, as detailed in Eq. 9.

$$\mathcal{L}_{cls} = -CS([cls]_{I'}, (1 - \alpha)[cls]_I + \alpha[cls]_R) \tag{9}$$

Locally, the selected patches for semantic injection contain minimal information and thus have lower attention weights. To ensure the vision encoder captures the injected semantics, we reallocate attention weights of these patches by assigning them the scaled attention weights of the reference image $R$. Other patches undergo a similar scaling to preserve global semantic consistency. The process is described as Eq. 10. $A_R$ is the attention weights of $R$ and $\beta$ is the scale factor.

$$A'_i = \left\{ \begin{array}{ll} \beta A_{R_i}, & \text{if } mask_i = 0 \\ (1 - \beta)A_{I_i}, & \text{if } mask_i = 1 \end{array} \right\} \tag{10}$$

After obtaining the new attention weight map, we compute the key features by taking the product of the weight map and the patch token embeddings of $R$. Subsequently, we align the key features at corresponding positions in the adversarial example with those in $R$ to achieve local semantic injection, as shown in Eq. 11.

$$\mathcal{L}_p = -(1 - mask) * CS(A_{I'} * [patch]_{I'}, A' * [patch]_R) \tag{11}$$

Then we also consider to preserve the key features in other regions of the original image. Note that to allocate sufficient attention weights for injected semantics, we should utilize the reallocated attention weights $A'$. So the attention-$[patch]$ fixation can be expressed as Eq. 12.

$$\mathcal{L}_{fix} = -mask * CS(A_{I'} * [patch]_{I'}, A' * [patch]_I) \tag{12}$$

In conclusion, the overall optimization of adversarial examples is the same as Eq. 8. The pipeline of ADVEDM-A is shown in Fig. 3.

## 5 Experiment

We conducted experiments in **both general evaluation scenarios and EDM tasks**. The general scenario involves image description, which is consistent with existing adversarial attack evaluation scenarios in general VLMs. The EDM tasks include autonomous driving and robotic arm manipulation. Besides, more visualization results are provided in Appendix B and our webpage. The ablation studies and exploration of transferability also can be found in Appendix C and D.

Table 2: Quantitative results of attacks on Dolphins Benchmark dataset in autonomous driving scenes.

| Models
Attacks | LV-v2 | | MGPT-4 | | Otter | | Dol | | Avg. | |
|---|---|---|---|---|---|---|---|---|---|---|
| | ASR(%) | SS | ASR(%) | SS | ASR(%) | SS | ASR(%) | SS | ASR(%) | SS |
| PGD | 22.0 | 0.336 | 16.0 | 0.206 | 13.0 | 0.279 | 18.0 | 0.259 | 17.3 | 0.270 |
| MF-it | 17.0 | 0.303 | 21.0 | 0.250 | 15.0 | 0.253 | 13.0 | 0.237 | 16.5 | 0.261 |
| MF-ii | 24.0 | 0.315 | 17.0 | 0.223 | 8.00 | 0.260 | 10.0 | 0.234 | 14.8 | 0.258 |
| ADVEDM-R | **74.0** | **0.511** | **78.0** | **0.505** | **62.0** | **0.485** | **84.0** | **0.502** | **74.5** | **0.501** |
| ADVEDM-A | **68.0** | **0.564** | **73.0** | **0.489** | **66.0** | **0.509** | **79.0** | **0.546** | **71.5** | **0.527** |

## 5.1 Setups

**Models.** We employ several commonly-used VLMs, including BLIP-2 [54], MiniGPT-4 (MGPT-4) [48], LLaVA-v2 (LV-v2) [47], Otter-Image (Otter-I) [55] and OpenFlamingo-v2 (OFLMG-v2) [56]. The vision-text encoders of BLIP-2 and MiniGPT-4 are based on EVA CLIP [26], while others are based on OpenAI's ViT CLIP [25].

**Datasets.** For general scenarios, we select MS-COCO 2014 [57, 58]. For the autonomous driving scenario, we choose Dolphins Benchmark [21] and DriveLM-nuScenes [4] that are specialized for this task, while for the robotic arm manipulation task, we sample 100 images from the physical world and construct instructions and actions.

**Attacks.** We choose several typical adversarial attacks as baseline, including CLIP-Based PGD [19], MF-it and MF-ii [10]. The norm constraint of the adversarial perturbation $\epsilon$ is set to 8/255. For CLIP-Based PGD, we set the number of iterations to 20 with a step size of 0.01. For MF-it and MF-ii, we use their official open-source code. Details of our methods' settings are in Appendix A.

**Metrics.** We define Attack Success Rate (ASR) and Semantic Similarity (SS) to measure attack effectiveness. Specifically, SS is the cosine similarity between output embeddings of adversarial and clean samples calculated by a text encoder. We introduce Semantic Preservation Rate (SPR), which measures the retention of other objects' semantics in VLM description of input images. The detailed formal definitions of these metrics are provided in Appendix A.

Note that the interpretation of evaluation metrics vary across different tasks. In the general image description task, an attack is considered successful if it causes the target object to include or exclude in the generated description. In this case, a higher ASR values indicate greater attack effectiveness, while higher SPR and SS values reveal the attack better preserves the semantic integrity of the image. In the EDM tasks, the attack is successful if the output decisions and plannings align with the expected results after the addition or removal of target object's semantics (*i.e.*, valid yet incorrect). However, since VLM-based EDM systems do not provide detailed descriptions of all objects in the inputs, SPR cannot be reliably computed and is thus omitted from the corresponding experiments.

## 5.2 Evaluation in General Scenarios

**Settings.** According to our threat model, we employ the vision-text encoder of each victim model as surrogate model to launch attack. For the attack target, we randomly select an object in the image to remove its semantics or choose an object not in the image to inject its semantics. We randomly select 1000 images to generate adversarial examples and record the average ASR, SPR and SS.

**Results.** Tab. 1 shows that though existing attacks achieve high ASR values, they suffer from low semantic preservation, with SPR values under 20%. This indicates they disrupt the majority of the original image's semantics, resulting in poor fine-grained control. In contrast, our methods exhibit a slight decrease in ASR, but their SPR values are significantly higher (66.7% and 71.6%, respectively), preserving most of the original image's semantics and enabling fine-grained control. Our methods also achieve higher SS values, as it preserves most of the original image's semantics.

## 5.3 Evaluation in EDM tasks

### 5.3.1 Autonomous Driving Task

**Settings.** We conduct experiments on two specialized datasets, Dolphins Benchmark and DriveLM-nuScenes, with three general VLMs (LLAVA-v2, MniGPT-4, and Otter-Image) and Dolphins (Dol) [21], a VLM designed for decision-making task in autonomous driving. For each dataset, we randomly

Table 3: Quantitative results of attacks on DriveLM-nuScenes dataset in autonomous driving scenes.

| Models Attacks | LV-v2 ASR(%) | SS | MGPT-4 ASR(%) | SS | Otter ASR(%) | SS | Dol ASR(%) | SS | Avg. ASR(%) | SS |
|---|---|---|---|---|---|---|---|---|---|---|
| PGD | 20.0 | 0.274 | 19.0 | 0.218 | 16.0 | 0.286 | 23.0 | 0.243 | 19.5 | 0.255 |
| MF-it | 19.0 | 0.263 | 24.0 | 0.242 | 9.00 | 0.245 | 17.0 | 0.237 | 17.3 | 0.247 |
| MF-ii | 14.0 | 0.260 | 17.0 | 0.221 | 12.0 | 0.253 | 14.0 | 0.234 | 14.3 | 0.242 |
| ADVEDM-R | **79.0** | **0.487** | **83.0** | **0.504** | **75.0** | **0.472** | **86.0** | **0.544** | **80.8** | **0.502** |
| ADVEDM-A | **73.0** | **0.536** | **79.0** | **0.517** | **71.0** | **0.509** | **82.0** | **0.530** | **76.3** | **0.523** |

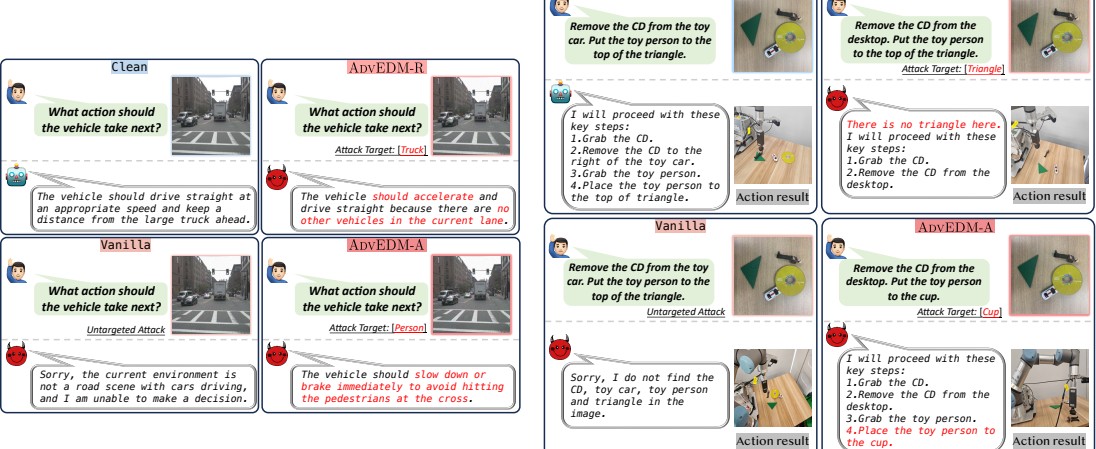

Figure 5: Visualization results in the autonomous driving decision-making task.

Figure 6: Visualization results in the VLM-based robotic arm manipulation task.

select 100 road scene images and choose common objects in the road as target, such as vehicles, pedestrians, and traffic lights. The impact of various attacks on the reasoning process of VLMs is presented in detail in Appendix C.

**Results.** The quantitative results are reported in Tab. 2 and 3, and the visualization results are shown in Fig. 5. According to the quantitative results, the average ASR of our methods are over 70% and 75% on the two datasets, respectively, dramatically outperforming existing attack methods. This demonstrates that our attack can have a substantial impact on VLM-based decision-making systems in autonomous driving. Our methods also achieve higher SS, as they ensure the model outputs valid decisions that are structurally identical to normal outputs, such as "The vehicle should go straight/turn left." Besides, the visualization results also demonstrate that our methods enable VLMs to make valid but incorrect decisions, thereby affecting the vehicle's driving state.

#### 5.3.2 Robotic Manipulation Task

**Settings.** We take the robotic arm manipulation task as an example, where various objects are placed on desktop, and then 100 images are captured to form the dataset. We still select three general VLMs and EmbodiedGPT (EmGPT) [24], a VLM specifically designed for this task, as victim models. For each image, the instructions we design involve the manipulation of two or more objects, including the target object selected for attack. Moreover, to better visualize the results, we deploy the VLMs on a UR robotic arm, where their outputs directly influence the robotic arm's actions. The impact of various attacks on the reasoning process of VLMs is also discussed in Appendix C.

**Results.** The quantitative results are shown in Tab. 4 and the visualization results are shown in Fig. 6. The presentation videos of attacking on robotic arm manipulation can be found on our website. The results highlight the superior effectiveness of our attack methods, achieving ASR values of 68.8% and 64.5%, significantly surpassing those of existing attacks. When VLMs encounter existing attacks, the perceived semantics are completely different from the original image, leading to error messages such as "Sorry, there is no object A nor B on the desktop.", resulting in lower SS values. In contrast,

Table 4: Quantitative results of attacks in robotic arm manipulation task.

| Attacks \ Models | LV-v2 | | MGPT-4 | | Otter | | EmGPT | | Avg. | |
|---|---|---|---|---|---|---|---|---|---|---|
| | ASR(%) | SS | ASR(%) | SS | ASR(%) | SS | ASR(%) | SS | ASR(%) | SS |
| PGD | 16.0 | 0.123 | 12.0 | 0.165 | 11.0 | 0.103 | 7.00 | 0.079 | 11.5 | 0.117 |
| MF-it | 11.0 | 0.132 | 15.0 | 0.185 | 14.0 | 0.118 | 10.0 | 0.090 | 12.5 | 0.131 |
| MF-ii | 9.00 | 0.119 | 12.0 | 0.155 | 6.00 | 0.091 | 9.00 | 0.095 | 9.0 | 0.115 |
| ADVEDM-R | **72.0** | **0.459** | **66.0** | **0.428** | **63.0** | **0.466** | **74.0** | **0.450** | **68.8** | **0.451** |
| ADVEDM-A | **67.0** | **0.417** | **63.0** | **0.448** | **58.0** | **0.436** | **70.0** | **0.487** | **64.5** | **0.447** |

our methods mainly alter the VLM's decisions and plannings regarding the target object, while the decisions and plannings for non-target objects remain similar to those make for clean images, ensuring that the SS values remain sufficiently higher.

# 6 Conclusion

In this work, we propose a fine-grained adversarial attack framework ADVEDM against the VLM-based EDM systems, which aims to reveal the vulnerabilities of them. By only disrupting the VLM's perception of the target object while preserving other semantics, our attack maintains the integrity of VLM's reasoning process, enabling it to output valid yet incorrect decisions that influence the entity's interactions with the real world. Specifically, we design two attack variants: ADVEDM-R, which removes the semantics of a specific object from the image, and ADVEDM-A, which injects the semantics of a new object into the image. Experimental results in both general scenarios and EDM tasks demonstrate the superior fine-grained control and attack effectiveness of our methods against the VLM-based EDM systems, outperforming existing attacks targeting the VLM itself.

## Acknowledgements

Minghui Li's work is supported by the National Natural Science Foundation of China under Grant No. 62202186. Shengshan Hu's work is supported by the National Natural Science Foundation of China under Grant No.62372196. Minghui Li is the corresponding author.

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

# A  Detailed Experimental Settings

**Parameter Settings of our methods.** For ADVEDM-R, we select the top 20% of image patches with the highest similarity to the target object's text embedding and mask their pixels to generate masked image $M$. We set $w_1$, $w_2$ and $w_3$ in Eq. (3) to 0.5, 2, and 0.2, respectively. The optimization is performed for 500 iterations using the Adam optimizer [59] with a learning rate of 0.005. For ADVEDM-A, we We select a $100 \times 100$ pixel region in the image for semantic injection. When reallocating attention weights, we set the scaling factor $\beta = 0.4$ in Eq. (12), and the fusion weight $\alpha$ of [CLS] tokens is 0.5 in Eq. (11). During the optimization process, $w_1$ to $w_3$ is set to 0.8, 2, and 0.3, respectively. The remaining optimization settings are kept identical to those of ADVEDM-R.

**Experimental environment.** All experiments are conducted on NVIDIA A100-SXM4 GPUs, each equipped with 80GB of memory. For the calculation of metrics ASR and SPR, we adopt the LLM-as-judge approach [45], employing GPT-3.5-turbo and other LLMs.

**Definition of metrics.** Semantic Similarity (SS) is defined as the cosine similarity of between output embeddings of adversarial and clean images calculated by a text encode, which can be expressed as Eq. 13:

$$SS = CS(E_t(T), E_t(T'))$$  (13)

where $CS(\cdot)$ is the cosine similarity function and $T$ and $T'$ represent the textual outputs of clean images and adversarial examples respectively.

As for calculating SPR, the procedure is divided into two steps. The first step involves decomposing all objects included in the description $T$ and $T'$, which can be done by GPT-3.5-turbo, as shown in Eq. 14. The specific prompts for extracting the semantics of objects are provided in our source code.

$$\mathbf{D} = L(T) = \{Obj_1, Obj_2, ..., Obj_n\}$$
$$\mathbf{D'} = L(T') = \{Obj'_1, Obj'_2, ..., Obj'_{n'}\}$$  (14)

$L$ is the LLM served as the judge. $\mathbf{D}$ and $\mathbf{D'}$ are the set of objects whose semantics are included in the description $T$ and $T'$. Then we compute the SPR value as the preservation rate of the original object semantics in $\mathbf{D'}$, expressed as Eq. 15.

$$SPR = \frac{|\mathbf{D'} \bigcap \mathbf{D}|}{|\mathbf{D}|}$$  (15)

where $|\cdot|$ here represents the number of elements in the set.

# B  More Results of Experiments in EDM tasks

Here, we illustrate the reasoning process of VLM-based embodied decision-making systems to better demonstrate how our methods precisely affect the VLM's CoT while preserving its structural integrity.

The CoT of VLMs in the autonomous driving task under various attacks as follows. The input image and prompts are identical to Fig. 5 in the manuscript. Our methods enable the VLM to correctly reason about non-target objects (like the green traffic lights) while inducing incorrect reasoning specifically for the target object. This demonstrates the fine-grained control of our methods, accurately manipulating targeted part of the VLM's reasoning process.

> **CoT of the clean image:** *In current scenario, the green light allows the vehicle to proceed through the intersection. Given the surrounding traffic—specifically, a truck ahead in the current lane and cars in the left lane—the vehicle should maintain a safe speed and distance while continuing straight.*

> **CoT of existing attacks:** *The current scene depicts a plate of fruit on a table with a cat nearby. As it bears no clear connection to a road scenario, no relevant decision can be made.*

> **CoT of ADVEDM-R:** *In the current scenario, the traffic light is green, allowing the vehicle to proceed straight. With no vehicles or pedestrians ahead in the lane, it can accelerate safely.*

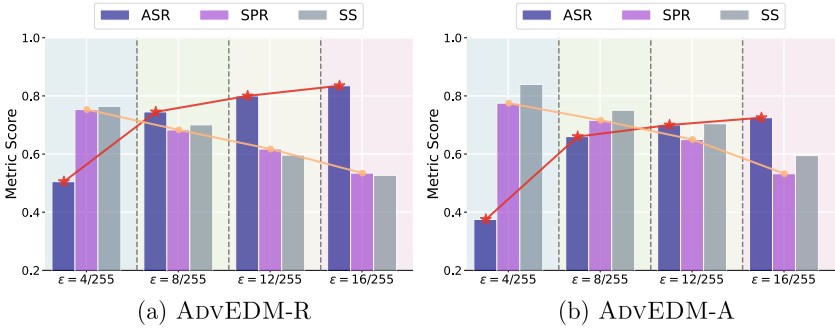

(a) ADVEDM-R                  (b) ADVEDM-A

Figure 7: Results of our methods under various $\epsilon$ values.

> **CoT of ADVEDM-A:** *In the current scenario, the green traffic light indicates that the vehicle can proceed straight. However, since a pedestrian is at the intersection, the vehicle should slow down and brake to yield. Once the pedestrian has passed, it can continue forward.*

The CoT of VLMs in robotic arm manipulation are as follows. The input image and prompts are consistent with Fig. 6 in the manuscript. Existing attacks prevent the VLM from recognizing any objects mentioned in the instructions, disrupting the reasoning process and terminating the task. ADVEDM-R enables the VLM to remain unaware of the target object while leaving other objects unaffected. As a result, the system makes incorrect decisions related to the target object while correctly executing other instructions. ADVEDM-A makes the VLM perceive the injected semantics of the target object, and then make corresponding decisions and plannings for instructions involving the object.

In conclusion, our methods perturb the VLM's perception process to precisely manipulate its reasoning about the target object while maintaining the coherence and integrity of the entire CoT.

> **CoT of the clean image:** *Seeing a CD on the toy car, grasp and remove it. Noticing the toy person and the triangle, pick up the toy person and carefully place it on top of the triangle, ensuring stability.*

> **CoT of existing attacks:** *Detected an apple and a book on the table, but no CD, toy car, or triangle. Adjust the camera to search further. Found a fork and a doll in the corner, but no matching objects. Task aborted.*

> **CoT of ADVEDM-R:** *Detected a CD, a toy car, and a toy person but no triangle. Remove the CD first. With no target for the toy person, pause the task and await further instructions.*

> **CoT of ADVEDM-A:** *Detected a CD, a toy car, a toy person, a triangle, and a cup. First, grab and remove the CD. Then, seeing the cup as the target container, decide to pick up the toy person and carefully place it into the cup, ensuring stability before completing the task.*

## C   Ablation Studies

**Influence of $\epsilon$.** The $\epsilon$ in the manuscript is the norm constraint of adversarial perturbation. Here, we set a range of $\epsilon$ values and select 100 images from MS-COCO for evaluation. Other settings are the same as Sec. 5.1 in the manuscript. The average results for various metrics in five victim VLMs are shown in Fig. 7. The visualization results under different $\epsilon$ are also illustrated in Fig. 8.

According to the results, the attack effectiveness is inferior when $\epsilon$ is set to 4/255, as the adversarial perturbation is too slight to disrupt the perception of VLMs. As $\epsilon$ increases, the magnitude of adversarial perturbation grows and our methods achieve stronger attack effectiveness. However, too large magnitude of perturbation breaks the semantics of non-target objects in the image, resulting in a significant drop in the SPR values, especially when $\epsilon \geq 12/255$. So $\epsilon = 8/255$ serves as an optimal setting that balances attack effectiveness and fine-grained control.

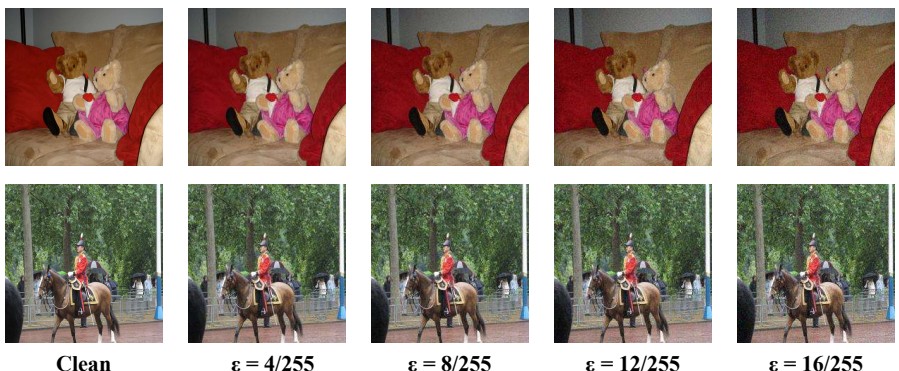

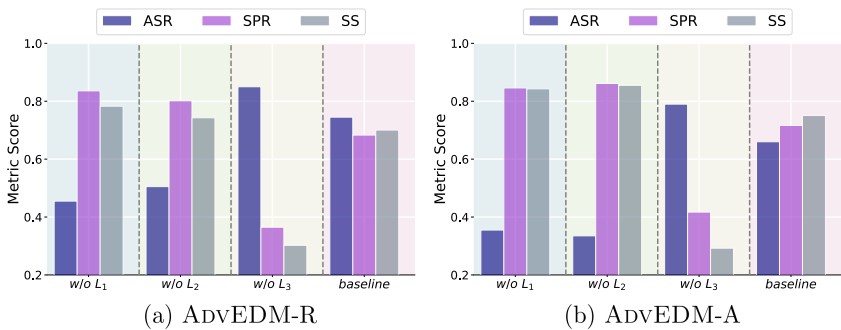

| Clean | ε = 4/255 | ε = 8/255 | ε = 12/255 | ε = 16/255 |

Figure 8: Visualization results under different $\epsilon$ settings.

(a) ADVEDM-R

(b) ADVEDM-A

Figure 9: Results of ablation study about three loss functions of our methods.

**Ablation about loss functions.** As shown in Eq. (10) in the manuscript, the optimization objective of our methods consists of three loss functions: $L_{cls}$, $L_p$, and $L_{fix}$. We also select 100 images from MS-COCO and also keep other settings identical to Sec. 5.2. The average results of various metrics are illustrated in Fig. 9.

The ASR values decrease dramatically without $L_{cls}$ or $L_p$, which demonstrates that removing or injecting the target object's semantics from both global and local perspectives is more effective. Additionally, $L_{fix}$ is critical to preserve non-target objects' semantics, as the SPR values decrease to about 0.35 and 0.4 respectively without $L_{fix}$.

**Ablation about modules and hyper-parameters in ADVEDM-A.** As for ADVEDM-A, we additionally provide some experimental results of ablation studies on attention-weight reallocation and the hyper-parameters $\alpha$ and $\beta$ in Eq. 9 and 10 to investigate their impact on the attack performance. Specifically, we randomly select 100 images from the COCO dataset and evaluate the results on MiniGPT-4 and LLAVA-V2. All other experimental settings remain the same as those described in Appendix A. The results are presented in Tab. 5 and 6, respectively.

According to Tab. 5, without the attention-weight reallocation, both SPR and SS drop significantly, which highlights the importance of this mechanism for fine-grained control. It ensures sufficient attention is allocated to the original regions of the image, enabling the model to retain awareness of the remaining semantics after the new semantic injection.

The results in Tab. 5 and 6 indicate that both $\alpha$ and $\beta$ influence the strength of semantic injection for the target objects. Larger values of them lead to increased attention weights assigned to the injected semantics, resulting in higher ASR. However, this also affects the model's perception of the semantics of other objects in the image, leading to a decrease in SPR and SS. Overall, to better balance the two aspects, we adopt $\alpha = 0.5$ and $\beta = 0.4$ as our default configuration.

Table 5: Ablation study of reallocation module and $\alpha$ in Eq. 9.

| Settings | ASR (%) | SPR (%) | SS |
|---|---|---|---|
| $\alpha = 0.3$ | 58.0 | 85.3 | 0.804 |
| $\alpha = 0.4$ | 67.5 | 80.7 | 0.776 |
| $\alpha = 0.5$ | **72.5** | **74.8** | **0.727** |
| $\alpha = 0.6$ | 78.0 | 66.1 | 0.655 |
| $\alpha = 0.7$ | 81.5 | 53.6 | 0.583 |
| $w/o$ realloc. | 70.5 | 44.3 | 0.515 |

Table 6: Ablation study of $\beta$ in Eq. 10.

| Settings | ASR (%) | SPR (%) | SS |
|---|---|---|---|
| $\beta = 0.2$ | 64.0 | 81.6 | 0.785 |
| $\beta = 0.3$ | 69.5 | 79.0 | 0.752 |
| $\beta = 0.4$ | **72.5** | **74.8** | **0.727** |
| $\beta = 0.5$ | 74.0 | 67.6 | 0.690 |
| $\beta = 0.6$ | 77.0 | 59.3 | 0.644 |

Table 7: Ablation studies of the weights of loss functions in ADVEDM-R and ADVEDM-A.

| Metrics / Settings | ADVEDM-R | | | ADVEDM-A | | |
|---|---|---|---|---|---|---|
| | ASR (%) | SPR (%) | SS | ASR (%) | SPR (%) | SS |
| group (1) | 79.5 | 64.3 | 0.626 | 77.0 | 69.2 | 0.674 |
| group (2) | 82.0 | 61.8 | 0.609 | 81.5 | 63.1 | 0.658 |
| group (3) | 69.0 | 74.5 | 0.723 | 65.5 | 79.6 | 0.771 |
| baseline | 75.0 | 71.8 | 0.691 | 72.5 | 74.8 | 0.727 |

**Ablation about the weights of loss functions.** We conduct studies on various combinations of loss function weights, using the same data and model settings as described above. For AdvEDM-R, we set the groups: (1) $w_1 = 0.8, w_2 = 2, w_3 = 0.2$; (2) $w_1 = 0.5, w_2 = 2.3, w_3 = 0.2$; (3) $w_1 = 0.5, w_2 = 2.0, w_3 = 0.5$; (4) $w_1 = 0.5, w_2 = 2.0, w_3 = 0.2$ (baseline). For AdvEDM-A, we set the groups: (1) $w_1 = 1.0, w_2 = 2.0, w_3 = 0.3$; (2) $w_1 = 1.0, w_2 = 2.2, w_3 = 0.3$; (3) $w_1 = 0.8, w_2 = 2.0, w_3 = 0.5$; (4) $w_1 = 0.8, w_2 = 2.0, w_3 = 0.3$ (baseline). The results are presented in Tab. 7.

According to the quantitative results, the weights $w_1$ and $w_2$ control the removal and injection of target semantics. Increasing their weights tends to improve the ASR, as more attention is directed toward the semantics of the target object. However, this also reduces the preservation of the original semantics in the image, leading to lower SPR and SS values. By contrast, increasing $w_3$ encourages the model to preserve more of the original semantics, but it weakens the target objects' semantics, resulting in a decrease in ASR.

## D Exploration of Transferability

To further evaluate the performance of our methods in black-box scenarios where the adversary has no knowledge about the victim model, we adapt them into transfer-based attacks. Specifically, we adopt SSA-CWA algorithm [60, 61] during the optimization process and employ four vision-text encoders, CLIP-ViT-L14, CLIP-ViT-B32, CLIP-ViT-bigG-14, and ViT-SO400M-14-SigLIP [62], as an ensemble of surrogate models.

**Settings.** We randomly select 100 images from MS-COCO and set target objects. The victim models include four commercial black-box VLMs: GPT-4o [63], Gemini-2.0 [2], Claude 3.5 [64]. The number of iterations is set to 30 for SSA-CWA, and the constraint of perturbation $\epsilon$ is 16/255. Other settings are identical to Sec. 5.2.

**Results.** The results are shown in Tab. 8. In the more challenging black-box setting, the attack effectiveness of our methods degrades, with the ASR of the two methods dropping by approximately 20% and 30% compared with attacks in the gray-box setting. Nevertheless, our methods still maintain a notable level of effectiveness against commercial VLMs while preserving fine-grained control, highlighting their potential for transferability to black-box scenarios. How to further enhance the transferability of our attacks will be explored in future work.

Table 8: Results of our methods in the black-box setting.

| ADVEDM-R | | | | |
|---|---|---|---|---|
| Metrics \ Models | GPT-4o | Gemini | Claude | Average |
| ASR(%) | 58.0 | 55.0 | 60.0 | 57.7 |
| SPR(%) | 71.3 | 68.1 | 70.6 | 70.0 |
| SS | 0.656 | 0.631 | 0.611 | 0.632 |

| ADVEDM-A | | | | |
|---|---|---|---|---|
| Metrics \ Models | GPT-4o | Gemini | Claude | Average |
| ASR(%) | 44.0 | 38.0 | 35.0 | 39.0 |
| SPR(%) | 73.7 | 76.4 | 77.5 | 75.9 |
| SS | 0.693 | 0.701 | 0.718 | 0.704 |

# E    Discussion about Limitations

In this work, we focus on leveraging adversarial perturbations to achieve fine-grained control over a specific aspect of image semantics, namely the presence or absence of a particular object. In fact, other levels of fine-grained semantics, such as altering the spatial location of objects or their interrelations, may also induce valid yet incorrect decisions in embodied EDM systems. A comprehensive analysis of these dimensions remains beyond the scope of this work, and we will further explore these aspects in our future work.

Moreover, our attack involves manipulating the image uploaded by the user to the VLM, thereby interfering with the system's decision-making process in the digital domain. While certain existing techniques, such as network interception and packet tampering—could potentially enable such attack, it offers limited flexibility. In future work, we plan to explore physically deployable adversarial examples (*e.g.*, adversarial patches) to enable more passive and practical attack scenarios in the physical world.

