# OpenReview forum: "AdvEDM: Fine-grained Adversarial Attack against VLM-based Embodied Agents"
_NeurIPS.cc/2025/Conference — NeurIPS 2025 poster_

### Official Review · Reviewer_UBmf · 2025-06-16

**Clarity:** 3
**Significance:** 4
**Originality:** 4
**Rating:** 5
**Confidence:** 4

**Summary:**

This paper explores adversarial attacks on VLM-based embodied decision-making systems by manipulating object-level semantics in visual feature maps. The method generates masks to insert or remove patches based on semantic similarity with target labels, and applies a feature-level alignment loss to alter masked regions while preserving context. Experiments on image captioning and autonomous driving demonstrate the method’s effectiveness. Transferability to proprietary LLMs such as GPT-4o, Gemini, and Claude is also well explored.

**Questions:**

See the weakness above

**Ethical Concerns:**

["NO or VERY MINOR ethics concerns only"]

**Final Justification:**

My concerns are all addressed. Based on the reviews and rebuttals, I lean towards acceptance for this work.

**Limitations:**

See the weakness above

**Quality:**

4

**Strengths And Weaknesses:**

Strength

1.	This work introduces a novel concept of fine-grained adversarial attacks and demonstrates that increasing the level of granularity can lead to more effective manipulation of embodied EDM systems, with physical impacts on real-world interactions.
2.	The authors conduct experiments across diverse task settings, including a general image description task and two embodied decision-making scenarios grounded in the physical world—autonomous driving and robotic manipulation. The results validate the effectiveness of the proposed attack method in both digital and real-world scenarios.
3.	The authors’ demo webpage and video further vividly illustrate the effectiveness of the fine-grained attacks.


Weakness

1.	There are some typos. In Fig. 3 (ADVEDM-A part), the $A’ * [patch] _ R$ should be corrected as $A _ {I’} * [patch]_R$. In line 236, $[cls]$ should be replaced as $[CLS]$.
2.	The authors do not explore the impact of the weighting of different loss functions on the attack performance, and a corresponding ablation study should be included to strengthen the evaluation.
3.	The paper lacks an evaluation of the proposed attack under common defense methods, such as image purification techniques (e.g., [a] and [b]). Including such analyses would offer a more comprehensive assessment of the method’s robustness. I hope the authors could provide further discussion in this regard.
4.	I hope the authors can release the dataset collected during their robotic manipulation experiments and to further expand it, thereby facilitating future research in this area.

[a] Nie W, Guo B, Huang Y, et al. Diffusion models for adversarial purification. arXiv preprint arXiv:2205.07460, 2022.

[b] Naseer M, Khan S, Hayat M, et al. A self-supervised approach for adversarial robustness. Proceedings of the IEEE/CVF conference on computer vision and pattern recognition. 2020: 262-271.

---

> ### Author Rebuttal · Authors · 2025-07-30
>
> ### Responses to Reviewer **UBmf**
>
> Thank you very much for your support and suggestions on our work! You can find  the responses to your questions as follows:
>
> **Q1: Discussion of defenses**
>
> **A1:** We randomly selected 100 images from the COCO dataset and performed attacks following the settings described in Appendix A. We then apply the NRP method [1] to purify these adversarial examples generated by AdvEDM-R and AdvEDM-A and evaluate the purified images on both MiniGPT-4 and LLAVA-V2. The average results of each metric are reported in Tab. R1.
>
> Tab. R1: The results of baselines and defenses.
>
> |    Method    | ASR(%) | SPR(%) |  SS   |
> | :----------: | :----: | :----: | :---: |
> | NRP (Remove) |   24   |  92.2  | 0.953 |
> |   AdvEDM-R   |   75   |  71.8  | 0.691 |
> |  NRP (Add)   |  28.5  |  89.6  | 0.940 |
> |   AdvEDM-A   |  72.5  |  74.8  | 0.727 |
>
> According to the results, although the defense method can significantly reduce the effectiveness of our attack, the ASR values still remain as high as over 20%, indicating that the defense is not sufficient to completely eliminate the impact of our adversarial examples on VLMs.
>
>
>
> **Q2: Ablation of loss function's weights**
>
> **A2:** We conduct some small-scale ablation studies on various combinations of loss function weights, using the same data and model settings as described above. For AdvEDM-R, we set the groups: (1) w1=0.8, w2=2, w3=0.2; (2)w1=0.5, w2=2.3, w3=0.2; (3)w1=0.5, w2=2, w3=0.5; (4) w1=0.5, w2=2, w3=0.2(baseline). For AdvEDM-A, we set  set the groups: (1) w1=1, w2=2, w3=0.3; (2)w1=1, w2=2.2, w3=0.3; (3)w1=0.8, w2=2, w3=0.5; (4) w1=0.8, w2=2, w3=0.3(baseline) The results are presented in Tab. R2 and R3.
>
> Tab. R2: The impact of loss function weights on AdvEDM-R.
>
> | AdvEDM-R  | ASR(%) | SPR(%) |  SS   |
> | :-------: | :----: | :----: | :---: |
> | baseline  |   75   |  71.8  | 0.691 |
> | group (1) |  79.5  |  64.3  | 0.626 |
> | group (2) |   82   |  61.8  | 0.609 |
> | group (3) |   69   |  74.5  | 0.723 |
>
> Tab. R3: The impact of loss function weights on AdvEDM-A.
>
> | AdvEDM-A  | ASR(%) | SPR(%) |  SS   |
> | :-------: | :----: | :----: | :---: |
> | baseline  |  72.5  |  74.8  | 0.727 |
> | group (1) |   77   |  69.2  | 0.674 |
> | group (2) |  81.5  |  63.1  | 0.658 |
> | group (3) |  65.5  |  79.6  | 0.771 |
>
> According to the quantitative results, the weights w1 and w2 control the removal and injection of target semantics. Increasing their weights tends to improve the ASR, as more attention is directed toward the semantics of the target object. However, this also reduces the preservation of the original semantics in the image, leading to lower SPR and SS values. By contrast, increasing w3 encourages the model to preserve more of the original semantics, but it weakens the target objects' semantics, resulting in a decrease in ASR.
>
>
>
> **Q3: Release of our dataset**
>
> **A3:** Yes, we are planning to release the dataset we collected for embodied robotic manipulation during our evaluation. Besides, we are currently expanding it to include a wider range of scenes and objects. The dataset will be released upon the publication of our paper.
>
>
>
> [1]  A self-supervised approach for adversarial robustness. CVPR 2020

---

> > ### Comment · Reviewer_UBmf · 2025-08-04
> >
> > Thanks for the rebuttal. My concerns are all addressed. Thus, I maintain my positive score for this work.

---

### Official Review · Reviewer_4Wzm · 2025-06-20

**Clarity:** 3
**Significance:** 2
**Originality:** 3
**Rating:** 4
**Confidence:** 3

**Summary:**

This paper focuses on fine-grained adversarial attacks of Vision-Language Models (VLM), specifically in embodied decision-making (EDM) tasks. It is established in the gray-box setting, where the attacker only has access to the image-text encoder of the VLM. The proposed attack framework, AdvEDM, contains two variants. The first one, AdvEDM-R, removes the object semantics from the patch embeddings both globally via the class token and locally via utilizing object masks. In this case, the VLM will ignore the target object. The second one, AdvEDM-A, injects object semantics into the patch embeddings of a specific area, following a similar pipeline as AdvEDM-R. It relies on a pre-trained text2image model for generating the target object image. Both variants rely on a pre-computed object mask, obtained via a similarity matrix between the patch embedding and text encoding. The authors evaluate the attack performance on 3 tasks: image description (general), autonomous driving, and robotic manipulation. The method shows effectiveness in the attack success rate, and preserves the image semantics, as shown by semantic similarity (SS) and semantic preservation rate (SPR).

**Questions:**

- How would the effect be if simply combining stronger adversarial attacks, such as targeted CW [1] or AutoPGD [2], with L_fix? According to the ablation study, the effects of L_fix and other losses are disentangled.
- Is it possible to automatically select the complex hyperparameters?

[1] Carlini, Nicholas, and David Wagner. "Towards evaluating the robustness of neural networks." 2017 ieee symposium on security and privacy (sp). Ieee, 2017.

[2] Croce, Francesco, and Matthias Hein. "Reliable evaluation of adversarial robustness with an ensemble of diverse parameter-free attacks." International conference on machine learning. PMLR, 2020.

Overall, my concerns are mainly on the complexity and inefficiency of AdvEDM. However, this is not strict, as the main claim is to perform targeted, fine-grained attacks and further reveal the vulnerability of VLM in physical-world EDM tasks.

**Ethical Concerns:**

["NO or VERY MINOR ethics concerns only"]

**Final Justification:**

The authors mention that there is computational overhead and it linearly increases with the number of iterations. This remains to be an issue as I mentioned in the Weaknessess part, but I do not think it's critical as claimed in my review. Therefore, I lean towards a borderline accept.

**Limitations:**

The authors discuss the limitations in Section E.
Negative societal impact: The authors mention the goal of revealing the vulnerability of the VLM in EDM tasks.

**Paper Formatting Concerns:**

The paper follows NeurIPS formatting.

**Quality:**

2

**Strengths And Weaknesses:**

### Strengths
- The proposed method is effective in achieving high ASR in EDM tasks, demonstrating its ability to alter the decision-making process in a targeted manner.
- The preservation of image semantics is quantified and validated using the proposed metrics, SS and SPR.
- The paper is well-written and easy to follow.

### Weaknesses

- According to the Appendix, the proposed method needs 500 iterations of optimization per image, which can lead to large computational overhead. Given the size of modern image encoders in VLMs, I'm concerned about the practicality (time consumption, memory requirement, etc) of the proposed method.
- Manual selection of the background patches in AdvEDM-A seems unrealistic in real-world scenarios.
- Reliance on an extra text2image model in AdvEDM-A undermines the practicality.
- No ablation studies on the numerical values (such as $\alpha, \beta,$ etc) used in the framework, albeit the ablation on setting the loss weights to 0. In other words, there is no discussion on the effects of these hyperparameters in the proposed framework.

---

> ### Author Rebuttal · Authors · 2025-07-30
>
> ### Responses to Reviewer **4Wzm**
>
> Thank you very much for your support and suggestions on our work! You can find  the responses to your questions as follows:
>
> **Q1: The computational overhead of our method**
>
> **A1:** Compared to the baselines used in our experiments, such as MF-it and MF-ii, our method is consistent with them in manipulating the image embedding to align with the target embedding. **Therefore, our methods do not introduce additional computational complexity.** Although our methods require more iterations to reach convergence, the overall time overhead increases only linearly relative to the baselines, which is acceptable.
>
> Furthermore, compared to existing fully white-box attacks such as [1] and [2], our methods do not need to access to the complex downstream LLMs. This significantly improves the efficiency of our attacks and reduces computational overhead, which can be also considered as one of the advantages.
>
>
>
> **Q2: Manual selection of the background patches in AdvEDM-A**
>
> **A2:** In fact, the region selection strategy used for semantic injection in AdvEDM-A is independent of our loss function design. We have considered employing segmentation models, such as OVSeg [3], to automate the region selection process. However, these models may include some small objects within background regions in the segmentation results, which can lead to unintended disruption of original semantics during injection. Therefore, to ensure better attack effectiveness and fine-grained control, we finally adopt manually selecting the target regions for semantic injection. But we are interested in exploring more automated and accurate region selection algorithms further in future work.
>
>
>
> **Q3: Combining stronger attack methods with L_fix**
>
> **A3:** This is a very interesting suggestion! Taking AdvEDM-R as an example, we apply the CW algorithm [4] to jointly optimize L_cls and L_p while remaining L_fix unchanged. We then generate 100 adversarial examples from COCO dataset and evaluate them on MiniGPT-4 and LLAVA-V2. The results are reported in Tab. R1.
>
> Tab. R1 The comparision of results between combining CW with L_fix and our baseline (AdvEDM-R).
>
> |   Method   | ASR(%) | SPR(%) |  SS   |
> | :--------: | :----: | :----: | :---: |
> | CW + L_fix |  84.5  |  57.4  | 0.506 |
> |  baseline  |   75   |  71.8  | 0.691 |
>
> According the results, employing a stronger attack algorithm can indeed improve ASR. But unfortunately, this also significantly weakens the influence of L_fix on the adversarial examples, making it less effective in preserving the original semantics of the image. As a result, both SPR and SS scores decrease.
>
>
>
> **Q4: More ablation studies about hyperparameters**
>
> **A4:** As for AdvEDM-A, we provide some experimental results of small-scale ablation studies on the parameters α and β to investigate their impact on the attack performance. Specifically, we randomly select 100 images from the COCO dataset and evaluate the results on MiniGPT-4 and LLAVA-V2. All other experimental settings remain the same as those described in Appendix A. The results are presented in Tab. R2 and R3, respectively.
>
> Tab. R2: The ablation study of α.
>
> |       Setting        |  ASR(%)  |  SPR(%)  |    SS     |
> | :------------------: | :------: | :------: | :-------: |
> |        α=0.3         |    58    |   85.3   |   0.804   |
> |        α=0.4         |   67.5   |   80.7   |   0.776   |
> | **α=0.5 (baseline)** | **72.5** | **74.8** | **0.727** |
> |        α=0.6         |    78    |   66.1   |   0.655   |
> |        α=0.7         |   81.5   |   53.6   |   0.583   |
>
> Tab. R3:  The ablation study of β.
>
> |       Setting        |  ASR(%)  |  SPR(%)  |    SS     |
> | :------------------: | :------: | :------: | :-------: |
> |        β=0.2         |    64    |   81.6   |   0.785   |
> |        β=0.3         |   69.5   |   79.0   |   0.752   |
> | **β=0.4 (baseline)** | **72.5** | **74.8** | **0.727** |
> |        β=0.5         |    74    |   67.6   |   0.690   |
> |        β=0.6         |    77    |   59.3   |   0.644   |
>
> The results indicate that both α and β influence the strength of semantic injection for the target objects. Larger values of them lead to increased attention weights assigned to the injected semantics, resulting in higher ASR. However, this also affects the model’s perception of the semantics of other objects in the image, leading to a decrease in SPR and SS. Overall, to better balance the two aspects, we adopt α=0.5 and β=0.4 as our parameter configuration. We will include these ablation studies in the revised version of the paper.
>
>
>
> [1] An image is worth 1000 lies: Adversarial transferability across prompts on vision-language models. ICLR 2024
>
> [2] Image Hijacks: Adversarial Images can Control Generative Models at Runtime. ICML 2024
>
> [3] Open-Vocabulary Semantic Segmentation With Mask-Adapted CLIP. CVPR 2023
>
> [4] Towards Evaluating the Robustness of Neural Networks. S&P 2017

---

> > ### Comment · Reviewer_4Wzm · 2025-08-02
> >
> > Thanks for your rebuttal and the new results. I decide to keep my score.

---

### Official Review · Reviewer_hQgb · 2025-06-27

**Clarity:** 2
**Significance:** 2
**Originality:** 2
**Rating:** 4
**Confidence:** 4

**Summary:**

This paper proposes ADVEDM, a fine-grained adversarial attack framework targeting vision-language model-based embodied decision-making systems. Unlike previous approaches that generate coarse global perturbations causing invalid outputs, ADVEDM aims to preserve the overall scene semantics while subtly manipulating the perception of specific objects and to achieve valid yet incorrect outputs.

**Questions:**

1. The paper uses cosine similarity between patch embeddings and object text embeddings for semantic region selection, which may be less reliable in complex, multi-object scenes. Have the authors considered comparing this with more robust alternatives such as segmentation or object grounding (e.g., GLIP)? A comparison or further justification would be helpful.
2. Could you elaborate on the rationale for requiring manual selection of contiguous patches in ADVEDM-A? Additionally, how does the method perform when applied to images with minimal background space or densely packed objects?
3. The attention reallocation is one of key components of ADVEDM-A, but its impact is not evaluated. Including an ablation study would help clarify its contribution to performance. Also, have you examined the sensitivity of other key hyperparameters (e.g., scaling factor β) to understand their influence on attack success or semantic preservation?
4. What target response templates are used to guide baseline attacks toward object injection or removal? For example, if the goal is to inject a “ruler to the left of a tape on the table,” how is this target specified for the baselines? Clarifying this would help assess the fairness of the comparisons.

**Ethical Concerns:**

["NO or VERY MINOR ethics concerns only"]

**Final Justification:**

The authors' rebuttal has addressed my concerns, so I increase my score to 4.

**Limitations:**

See weakness.

**Quality:**

2

**Strengths And Weaknesses:**

**Strengths:**
1. The motivation of the paper is well-grounded. Ensuring the robustness of VLM-based Embodied Decision-Making (EDM) systems is an important and timely challenge, particularly given the growing deployment of such systems in real-world applications like autonomous driving and robotic manipulation.
2. The paper introduces a fine-grained adversarial attack framework tailored to EDM systems, which is a meaningful shift from existing coarse-grained attacks that tend to disrupt global semantics.
3. This paper includes a black-box evaluation on commercial VLMs showing moderate success, demonstrating the broader applicability of the method.

**Weaknesses:**
1. The method relies heavily on cosine similarity between image patch embeddings and object text embeddings to determine semantic relevant regions. This assumption may not hold in cluttered or complex scenes. There exist more advanced region selection models (e.g., semantic segmentation or object grounding methods) that could provide stronger supervision or robustness. A comparison with such methods is missing.
2. The ADVEDM-A method requires manual selection of contiguous patches, usually from less semantically meaningful regions like the background, which may limit its automation and scalability. It also remains unclear how the method performs on naturally crowded or complex images where background space is limited.
3. The Semantic Preservation Rate (SPR) metric compares object sets before and after attack to evaluate preservation. However, it does not consider other semantic confusion—i.e., unintended additions of new semantics beyond the target. As a result, the SPR score may overestimate semantic preservation. A more fine-grained or robust metric may be necessary.
4. The proposed attention-weight reallocation mechanism (particularly in ADVEDM-A) is claimed to enhance the preservation of non-target semantics. However, the paper does not include an ablation study to isolate and quantify its actual impact on attack success or semantic preservation. Additionally, other hyperparameters such as the scaling factor β are not analyzed in terms of sensitivity or influence on performance.

---

> ### Author Rebuttal · Authors · 2025-07-30
>
> ### Responses to Reviewer **hQgb**
>
> Thanks very much for your insightful suggestions on our work! You can find  the responses to your questions as follows:
>
> **Q1: The region selection process of our method**
>
> **A1:** In fact, when designing the region selection strategy, we have considered using segmentation models such as open-vocabulary segmentation models. For AdvEDM-R, we here conduct a small-scale experiment where we randomly selected 100 images and determine the target regions using both OVSeg [1] and our cosine similarity-based method. We then compute the Intersection over Union (IoU) between the two sets of results, **yielding an average IoU of 0.836.** This demonstrates that our cosine similarity-based method also achieves a high level of accuracy in region selection. Due to the additional computational overhead introduced by external segmentation models, we ultimately adopted the cosine similarity-based method, which offers a more efficient alternative.
>
> As for AdvEDM-A, these segmentation models may include some small objects within background regions in the segmentation results, which can lead to unintended disruption of original semantics during injection. Therefore, to ensure better attack effectiveness and fine-grained control, we finally adopt manually selecting the target regions for semantic injection. But we are interested in exploring more automated and accurate region selection algorithms further in future work.
>
>
>
> **Q2: The effectiveness of our method on naturally crowded or complex images**
>
> **A2:** Thank the reviewer for the thoughtful consideration! This is indeed a limitation of our method. For such images, where semantic information is densely packed and there is insufficient available space or low-information patches (e.g., background regions) for injecting new semantics, it becomes challenging to fully preserve the original image content. But in future work, we plan to further explore this issue by designing a strategy that embeds the target semantics into existing semantic patches, enabling their coexistence.
>
>
>
> **Q3: The metrics for evaluating unintended additions of new semantics beyond the target**
>
> **A3:** Actually, **one of the evaluation metrics we adopt, semantic similarity (SS), incorporates the assessment of unintended semantics**. A higher SS value indicates a smaller semantic discrepancy between the adversarial example and the original image, implying fewer unintended semantics are introduced.
>  To more rigorously quantify this aspect, we here define a dedicated metric, *USR* (Unintended Semantic Rate), which is computed as follows:
> $$
> \frac{|D' - (D \cap D')| }{|D|}
> $$
> where the meaning of each symbol is consistent with Eq. (15) in Appendix A. Then we compute the USR for each method based on the experiments reported in Tab. 1 of the manuscript. The results are presented in Tab. R1.
>
> Tab. R1: The average USR values of each method.
>
> | Metrtic | PGD  | MF-it | MF-ii | AdvEDM-R | AdvEDM-A |
> | :-----: | ---- | ----- | ----- | -------- | -------- |
> | USR (%) | 18.3 | 22.9  | 26.2  | **9.06** | **7.52** |
>
> According to the result, the USR values of our methods are all below 10% and significantly lower than those of other baselines, which are also consistent with their SS values in Tab.1 in the manuscript. This indicates that, compared with existing attacks, our methods are less likely to introduce unintended new semantics, demonstrating a clear advantage in fine-grained control.
>
>
>
> **Q4: More ablation studies about attention-weight reallocation mechanism**
>
> **A4:** As for AdvEDM-A, we additionally provide some experimental results of small-scale ablation studies on attention-weight reallocation and the parameters α and β to investigate their impact on the attack performance. Specifically, we randomly select 100 images from the COCO dataset and evaluate the results on MiniGPT-4 and LLAVA-V2. All other experimental settings remain the same as those described in Appendix A. The results are presented in Tab. R2, R3 and R4, respectively.
>
> Tab. R2: The ablation study of attention-weight reallocation module.
>
> |     Setting      | ASR(%) | SPR(%) | SS    |
> | :--------------: | :----: | :----: | ----- |
> | w/o reallocation |  70.5  |  44.3  | 0.515 |
> |     baseline     |  72.5  |  74.8  | 0.727 |
>
> Tab. R3: The ablation study of α.
>
> |       Setting        |  ASR(%)  |  SPR(%)  |    SS     |
> | :------------------: | :------: | :------: | :-------: |
> |        α=0.3         |    58    |   85.3   |   0.804   |
> |        α=0.4         |   67.5   |   80.7   |   0.776   |
> | **α=0.5 (baseline)** | **72.5** | **74.8** | **0.727** |
> |        α=0.6         |    78    |   66.1   |   0.655   |
> |        α=0.7         |   81.5   |   53.6   |   0.583   |
>
> Tab. R4:  The ablation study of β.
>
> |       Setting        |  ASR(%)  |  SPR(%)  |    SS     |
> | :------------------: | :------: | :------: | :-------: |
> |        β=0.2         |    64    |   81.6   |   0.785   |
> |        β=0.3         |   69.5   |   79.0   |   0.752   |
> | **β=0.4 (baseline)** | **72.5** | **74.8** | **0.727** |
> |        β=0.5         |    74    |   67.6   |   0.690   |
> |        β=0.6         |    77    |   59.3   |   0.644   |
>
> According to Tab. R2, without the attention-weight reallocation, both SPR and SS drop significantly, which highlights the importance of this mechanism for fine-grained control. It ensures sufficient attention is allocated to the original regions of the image, enabling the model to retain awareness of the remaining semantics after the new semantic injection.
>
> The results in Tab. R3 and R4 indicate that both α and β influence the strength of semantic injection for the target objects. Larger values of them lead to increased attention weights assigned to the injected semantics, resulting in higher ASR. However, this also affects the model’s perception of the semantics of other objects in the image, leading to a decrease in SPR and SS. Overall, to better balance the two aspects, we adopt α=0.5 and β=0.4 as our parameter configuration. We will include these ablation studies in the revised version of the paper.
>
>
>
> **Q5: The attack target settings for baseline attacks**
>
> **A4:** For the baseline methods, the attack target template is defined as either a text phrase or an image, following the settings in [2] and [3]. For example, when adding or removing the semantic concept “red apple", the target is specified as the text “red apple” or an image depicting it, and the attack is conducted by pulling the adversarial example closer to or pushing it away from the target in the embedding space. As for the more fine-grained semantic injection targets you mentioned, such as “ruler to the left of a tape on the table“, **such settings are not available and feasible for these baselines**, as they are unable to control the location or region of the targets.
>
>
>
> [1] Open-Vocabulary Semantic Segmentation With Mask-Adapted CLIP. CVPR 2023
>
> [2] On evaluating adversarial robustness of large vision-language models. NeurIPS 2023
>
> [3] On the robustness of large multimodal models against image adversarial attacks. CVPR 2024

---

> > ### Comment · Reviewer_hQgb · 2025-08-04
> >
> > Thank you for the detailed rebuttal and the additional experimental results. Your response effectively addresses several of my initial concerns.
> > However, some reservations remain. In particular, the proposed method relies on manual selection of contiguous image patches—typically from background regions or areas with minimal semantic content—for semantic injection. This manual process limits the degree of automation and may hinder scalability in more complex or cluttered visual environments. Additionally, the method appears less applicable in scenes with limited background, and it remains unclear how the availability and size of semantic injection regions affect attack success.

---

> ### Author Response · Authors · 2025-08-05
> **Responses to the Feedback of Reviewer hQgb**
>
> We appreciate your feedback on our rebuttal. In response to the reviewer's request for further investigation into how the availability and size of semantic injection regions affect attack success, we conducted additional experiments on LLAVA and MiniGPT4 still using 100 randomly selected COCO dataset images with three different region sizes (50×50, 75×75, and 100×100 pixels) while keeping all other experimental settings consistent with those above. The quantitative results are reported in Tab. R5.
>
> Tab. R5: Impact of sizes of selected region in AdvEDM-A
>
> |        Size        | ASR(%) | SPR(%) | SS    |
> | :----------------: | :----: | :----: | ----- |
> |       50*50        |   61   |  81.6  | 0.783 |
> |       75*75        |  68.5  |  77.1  | 0.754 |
> | 100*100 (baseline) |  72.5  |  74.8  | 0.727 |
>
> As shown in Tab. R5, our attack maintains over 60% ASR even when semantic injection regions are limited to relatively small areas (about 5% of the whole image, typically 50×50 pixels). **This demonstrates that our method is not highly sensitive to the size of selected background regions**. Additionally, we employ a segmentation model to perform semantic segmentation on 500 randomly selected images from the COCO dataset (covering a wide range of natural images) and the nuScenes dataset (specifically designed for real-world road images in autonomous driving task). The calculated average background proportions in these datasets are 75.48% and 64.93%, and **even the images with the smallest background proportions in these two datasets still reach 18.85% and 13.62% respectively, which are sufficient for our semantic injection**. Thus, our method can be effectively applied to these general scenarios.
>
> Furthermore, regarding the reviewer's concern about images with extremely dense and clustered foreground objects, injecting additional semantic objects would have less impact on the decision-making system, because the system’s attention is already saturated by existing objects. Instead, we recommend using AdvEDM-R to remove certain semantic elements from the image, as this approach would create a more pronounced effect on the system.
>
> In conclusion, **our method maintains attack performance even with small injection region sizes**, making it applicable to images with less background areas. We hope this would address your concern.

---

> ### Author Response · Authors · 2025-08-06
> **Follow up Our Response**
>
> Dear Reviewer hQgb,
>
> Thanks again for your feedback on our rebuttal! Regarding the issues you raised in your response, we have conducted additional experiments and provided further explanations, and we hope this will resolve your remaining concerns. With the discussion deadline approaching, we would greatly appreciate your further response.
>
> Sincerely,
>
> The Authors of Submission 16785

---

> ### Author Response · Authors · 2025-08-08
>
> Dear Reviewer hQgb,
>
> Thanks again for your valuable feedback! Regarding the issues you mentioned, we have provided additional experimental results and further explanations in our response above. Due to the discussion period ending in only one day, we want to gently check if our responses have addressed your remaining concerns.
>
> We would greatly appreciate your further response.
>
> Sincerely,
>
> The Authors of Submission 16785

---

### Official Review · Reviewer_DrxE · 2025-07-01

**Clarity:** 3
**Significance:** 3
**Originality:** 3
**Rating:** 3
**Confidence:** 4

**Summary:**

This paper proposes AdvEDM, a fine-grained adversarial attack on vision-language models (VLMs) in embodied decision making tasks. Unlike prior methods that rely on strong assumptions or disrupt global semantics, this approach perturbs only key objects to produce valid but incorrect decisions. Two variants—semantic removal and addition—demonstrate strong effectiveness across general and EDM specific benchmarks, revealing realistic safety risks from subtle, task-consistent attacks.

**Questions:**

See Weakness.

**Ethical Concerns:**

["NO or VERY MINOR ethics concerns only"]

**Final Justification:**

While the authors' response addresses some technical concerns, I remain worried about the application scenarios.

Embodied safety differs fundamentally from digital domain attacks and should focus on physically realizable threats like printable patches, rather than global perturbations. The authors propose network hijacking to add perturbations after image capture, but this assumption is logically flawed: if an attacker already possesses channel hijacking capabilities, this constitutes a far more severe security threat than adversarial attacks themselves, undermining the research premise. Additionally, the claimed transferability to patch attacks lacks intuitive justification and experimental validation.

If positioned as VLM adversarial attack research, the work shows technical innovation; however, when focused on embodied scenarios, its practical applicability remains questionable. Therefore, I maintain my original score.

**Limitations:**

The attack assumes access to images after capture by the robot’s camera, which is unrealistic in physical-world deployment where post-capture modification is impractical.

**Quality:**

3

**Strengths And Weaknesses:**

Strengths:

The paper addresses a timely and important topic—embodied safety in vision-language decision-making systems.

The motivation is clearly articulated: prior attacks often disrupt global semantics, whereas the proposed method subtly alters key objects while preserving overall scene context, posing more realistic and severe safety risks.

Weaknesses:

The proposed method appears tailored to decoupled VLM architectures and may not generalize to recent unified models such as OpenVLA series.

The set of baselines is limited and outdated; strong recent methods in the same setting, such as [1, 2], are missing.

The overall idea closely resembles [3], but the paper lacks adequate comparison and discussion.


[1]Chain of Attack: On the Robustness of Vision-Language Models Against Transfer-Based Adversarial Attacks, CVPR2025

[2]Efficient Generation of Targeted and Transferable Adversarial Examples for Vision-Language Models Via Diffusion Models, TIFS2024

[3] Replace-then-Perturb: Targeted Adversarial Attacks With Visual Reasoning for Vision-Language Models, arxiv 2024

---

> ### Author Rebuttal · Authors · 2025-07-30
>
> ### Responses to Reviewer **DrxE**
>
> Thank you very much for your valuable suggestions on our work! You can find  the responses to your questions as follows:
>
> **Q1: Comparison of latter baselines**
>
> **A1:** Recent works, including [1] and [2], primarily focus on enhancing the transferability and effectiveness of adversarial examples. Essentially, they can be seen as augmentations of MF-ii and MF-it, without explicitly addressing the design of refined loss functions to preserve the majority of the image's semantic content. As a result, these attacks lack fine-grained control.
>
> We also evaluated the effectiveness of these two attacks using their official implementations. The experiments were conducted on LLAVA-V2 and MiniGPT-4, with 100 randomly selected images from the COCO dataset. The average results of metrics are presented in Tab. R1.  These quantitative results indicate that, compared with our methods, although the attacks achieve relatively high ASR values, their SPR and SS scores remain low, which suggests that they lack fine-grained control.
>
> Tab. R1: Comparison of the results between two baselines and our methods.
>
> |       Method        | ASR(%) | SPR(%) |  SS   |
> | :-----------------: | :----: | :----: | :---: |
> |         COA         |   84   |  11.5  | 0.238 |
> |     AdvDiffVLM      |  76.5  |  15.2  | 0.273 |
> | **AdvEDM-R (Ours)** |   75   |  71.8  | 0.691 |
> | **AdvEDM-A (Ours)** |  72.5  |  74.8  | 0.727 |
>
>
>
> **Q2: Comparison with a similar work [3]**
>
> **A2:** The differences between our methods and the work [3] lie in the target region selection strategies and the fine-grained control of attacks. In the their work, the authors additionally rely on a CLIP segmentation model and a generative model, which introduces a strong assumption and extra computational cost. Moreover, their proposed method is limited to semantic replacement of objects in the image, lacking the flexibility to perform object-only removal or addition. By contrast, our methods require no external models and enables more precise and fine-grained adversarial attacks, including semantic removal and injection.
>
>
>
> **Q3: Concerns about the application and attack settings**
>
> **A3:** Most existing VLA models generate action sequences in an end-to-end manner **without involving reasoning or planning process, making it less accurate and explainable than VLMs**. So, in this work, we focus on the VLM-based embodied systems. However, recent works like [4] introduces a framework where a VLM is first used for reasoning and task planning, followed by a VLA model for action generation, thereby improving the accuracy of robotic manipulation. We plan to extend our attack to such models in embodied tasks in future work.
>
> Moreover, as for our attack settings, we follow the common practice in most existing works, such as [5] and [6], by adding adversarial perturbations to images captured by users in the digital domain, which is a widely adopted setting. In future work, we also plan to extend our attack to physical-world deployable adversarial patches to enhance its practicality.
>
>
>
> [1]Chain of Attack: On the Robustness of Vision-Language Models Against Transfer-Based Adversarial Attacks, CVPR2025
>
> [2]Efficient Generation of Targeted and Transferable Adversarial Examples for Vision-Language Models Via Diffusion Models, TIFS2024
>
> [3] Replace-then-Perturb: Targeted Adversarial Attacks With Visual Reasoning for Vision-Language Models, arxiv 2024
>
> [4] Hume: Introducing System-2 Thinking in Visual-Language-Action Model. arXiv 2025.
>
> [5] Visual adversarial attack on vision-language models for autonomous driving. arXiv 2024
>
> [6] Towards Trustworthy Autonomous Vehicles with Vision-Language Models Under Targeted and Untargeted Adversarial Attacks. CVPR 2025

---

> > ### Author Response · Authors · 2025-08-06
> >
> > Dear Reviewer DrxE,
> >
> > Thanks for your valuable review on our work! We have submitted a rebuttal response to address all of your comments. As the discussion period is approaching its deadline, we would like to kindly remind you. Your feedback would be greatly appreciated, as it is very important to us.
> >
> > Sincerely,
> >
> > The Authors of Submission 16785

---

> > ### Comment · Reviewer_DrxE · 2025-08-06
> >
> > Thank you for your response. My current concerns are primarily about the application of this method. On one hand, we would like to know whether this method has the potential to be transferred to architectures similar to OpenVLA. On the other hand, we have doubts about the practical scenarios of global perturbation attacks for embodied tasks that require interaction with the real world. This is because global perturbation attacks cannot be printed into the physical environment like unrestricted attacks or patch attacks. If the authors could provide more detailed explanations on these two points, I will consider raising the score.

---

> ### Author Response · Authors · 2025-08-06
> **Responses to the Feedback of Reviewer DrxE**
>
> Thanks for your valuable feedback.
>
> Regarding the first issue, as we mentioned above, existing VLAs (including OpenVLA) lack reasoning and planning processes, nor do they incorporate system prompts. Consequently, they cannot combine perceived images with specified backgrounds to determine task context consistency. These are just end-to-end models that generate corresponding action sequences for execution whenever target objects are perceived. **In other words, they exhibit poorer robustness and planning accuracy compared to VLM-based embodied AI systems.** Therefore, simple coarse-grained white-box and gray-box attacks remain effective against them [1]. **Since our method also disrupts VLA's perception, it intuitively achieves comparable attack effectiveness.**
>
> Besides, some latest works like [2] introduce a framework where a VLM is first used for reasoning and task planning, followed by a VLA model for action generation, thereby improving the reasoning and planning accuracy of the embodied system. We plan to extend our attack to such models in embodied tasks in future work.
>
>
>
> As for the second issue, actually, we follow most existing attack scenarios targeting embodied AI systems [3,4], which assume that the attacker obtains environment images captured by the system's sensors through means like network hijacking, then adds adversarial perturbations in the digital domain before feeding the perturbed images to the VLM for reasoning and planning.
>
> Regarding the physically deployable adversarial examples (e.g., printable patches) you mentioned, **we consider them orthogonal to our proposed fine-grained attack framework.** Based on our proposed loss functions of AdvEDM-R and AdvEDM-A, we can incorporate physical-world data augmentation techniques like Expectation Over Transformation (EOT), relax perturbation norm constraints, and reduce the number of optimized pixels during the optimization process. This will enable our attack framework to generate physically deployable adversarial patches, thereby enhancing their attack effectiveness in complex real-world environments. We are planning include these attempts in our future work.
>
>
>
> We hope the above explanations would resolve your concerns.
>
>
>
> [1] Exploring the adversarial vulnerabilities of vision-language-action models in robotics. ICCV 2025
>
> [2] Hume: Introducing System-2 Thinking in Visual-Language-Action Model. arXiv 2025
>
> [3] Visual adversarial attack on vision-language models for autonomous driving. arXiv 2024
>
> [4] Towards Trustworthy Autonomous Vehicles with Vision-Language Models Under Targeted and Untargeted Adversarial Attacks. CVPR 2025

---

> > ### Author Response · Authors · 2025-08-08
> > **Follow up Our Response**
> >
> > Thank you again for your insightful feedback. As for the application of our methods into VLAs and physical-world scenarios, we have provided further discussion and explanations in our response above. With the discussion period ending in only one day, we want to gently check if our responses have adequately addressed your concerns.
> >
> > We appreciate the opportunity for this dialogue and look forward to your reply.

---

### Note · Authors · 2025-08-12

Thanks all reviewers, ACs and SACs again for handling our submission #16785. In this work, we have proposed a novel adversarial attack framework against embodied AI safety in vision-language model based decision-making systems, which reveals the specific vulnerabilities of current embodied AI systems. Compared with existing adversarial attacks against VLMs, it can more precise fine-grained control and preserve the integrity of the reasoning and planning process of the system, thus making the embodied decision-making system output valid but incorrect  content and genuinely impacting the interaction between the embodied entities and the physical world. These contributions are widely acknowledged by all reviewers.



During the rebuttal and discussion period, our responses have resolved most of the reviewers' concerns. Specifically, as for the concerns about the application of our methods mentioned by Reviewer DrxE, we have provided further explanations and demonstrated the potential of our methods in applying into end-to-end VLA models and physical-world scenarios. Regarding the remaining concerns of Reviewer hQgb about "how the availability and size of semantic injection regions affect attack success", we have conducted additional experiments and provided further explanations to demonstrate that our method is not highly sensitive to the size of selected background regions and maintains attack performance with small injection region sizes, making it applicable to images with less background areas.



We hope our responses can address all of your concerns.

---

### Decision · Program_Chairs · 2025-09-17

**Decision:**

Accept (poster)

**Comment:**

The paper introduces a new attack methodology against embodied VLM-based systems that allows for "fine-grained" attacks --- that change the detection for only specific objects.

The paper received final scores of one accept, two borderline accepts, and one borderline reject. There is general agreement about the novelty / merit of the approach. The negative reviewer's main remaining concern is that the approach does not attempt to make the attacks physically realizable. Authors argue in return that physical reliazability is an orthogonal direction (i.e., the proposed approach could be combined with complementary research to make attacks physically realizable).  AC agrees with authors here, and further believes that non-physical attacks can be useful in themselves in assessing and improving the robustness of a system. Accordingly, the paper is recommended for acceptance.